# Role of Alternative Splicing in Regulating Host Response to Viral Infection

**DOI:** 10.3390/cells10071720

**Published:** 2021-07-08

**Authors:** Kuo-Chieh Liao, Mariano A. Garcia-Blanco

**Affiliations:** 1Genome Institute of Singapore, A*STAR, Singapore 138672, Singapore; 2Department of Biochemistry and Molecular Biology, University of Texas Medical Branch, Galveston, TX 77550, USA; 3Department of Internal Medicine, University of Texas Medical Branch, Galveston, TX 77550, USA; 4Institute of Human Infections and Immunity, University of Texas Medical Branch, Galveston, TX 77550, USA; 5Programme in Emerging Infectious Diseases, Duke-NUS Medical School, Singapore 169857, Singapore

**Keywords:** alternative splicing, antiviral response, innate immunity, cell death pathways

## Abstract

The importance of transcriptional regulation of host genes in innate immunity against viral infection has been widely recognized. More recently, post-transcriptional regulatory mechanisms have gained appreciation as an additional and important layer of regulation to fine-tune host immune responses. Here, we review the functional significance of alternative splicing in innate immune responses to viral infection. We describe how several central components of the Type I and III interferon pathways encode spliced isoforms to regulate IFN activation and function. Additionally, the functional roles of splicing factors and modulators in antiviral immunity are discussed. Lastly, we discuss how cell death pathways are regulated by alternative splicing as well as the potential role of this regulation on host immunity and viral infection. Altogether, these studies highlight the importance of RNA splicing in regulating host–virus interactions and suggest a role in downregulating antiviral innate immunity; this may be critical to prevent pathological inflammation.

## 1. Introduction

The host response to viral infection is multifaceted and incorporates the induction of an antiviral transcriptional program, including the expression of interferons (IFNs) and cytokines, and the activation of cell death pathways (apoptosis, necroptosis, and pyroptosis). Among these pathways, many steps are tightly regulated at several levels to ensure tissue homeostasis. In this review, we discuss the functional roles of alternative splicing and various spliced isoforms in shaping host immunity against viral infection.

Pre-messenger RNA splicing is an important RNA maturation step that involves the joining of exons and removal of introns. The overwhelming majority of transcripts produced by RNA polymerase II (RNAP II), including most mRNAs, contain introns and thus must be spliced. Splicing is carried out in cell nuclei by one of two macromolecular ribonucleoprotein complexes, known as the major and the minor spliceosomes [1]. It is estimated that more than 90% of expressed human genes undergo alternative splicing (AS) [2], which enables single genes to generate multiple distinct mRNAs that can encode distinct proteins, thus greatly expanding proteome complexity. Many types of AS events have been described, and they primarily include cassette exons, mutually exclusive exons, alternative 5′ splice site use, alternative 3′ splice site use, and intron retention. AS events can be regulated in a spatiotemporal-dependent manner [1] by the combined action of cis-elements (e.g., exon splicing enhancers (ESEs)) and trans-factors (e.g., RNA binding proteins) [3]. Aberrant splicing has been linked to many diseases [4,5], further underscoring the importance this highly regulated process. AS and mRNA isoforms play important roles in virtually all cellular processes and pathways, and it is thus not surprising that both have been noted to be critical for an effective antiviral response.

## 2. Alternative RNA Splicing and Its Isoforms in Type I and III IFN Responses

The antiviral response initiates when cellular pattern recognition receptors (PRRs) detect pathogen-associated molecular patterns (PAMPs). Cytosolic retinoic acid-inducible gene I (RIG-I) and melanoma differentiation-associated protein 5 (MDA-5) sense double-stranded RNA (dsRNA) (RIG-I specifically detects 5′-triphosphate or 5′-diphosphate of RNA molecules) and undergo conformational changes to associate with the downstream mitochondrial antiviral signaling protein (MAVS). Subsequently, MAVS associates with TANK Binding Kinase 1 (TBK1) and I-kappa-B kinase epsilon (IKKε), promoting the phosphorylation of interferon regulatory factor 3 (IRF3) and interferon regulatory factor 7 (IRF7). These two transcription factors translocate to the nucleus and drive the transcription and production of Type I and Type III interferon (IFN) mRNAs. The cytosolic DNA sensor, cyclic GMP-AMP synthase (cGAS), can detect DNA in the cytoplasm as PAMP, when DNA viruses infect cells and produce the cyclic dinucleotide 2′,3′-cyclic GMP–AMP (2′,3′-cGAMP) [6]. This secondary messenger in turn activates ER-resident stimulator of interferon genes (STING) and leads to TBK1-dependent IRF3 phosphorylation for Type I and III IFN production. It is important to note that cGAS has been shown to also respond to RNA virus infection, probably due to the cytoplasmic release of host mitochondrial DNA [7]. Additionally, membrane-bound Toll-like receptor 3 (TLR3) can recognize dsRNA in the endosomal compartments. Ligand detection by TLR3 triggers its association with the TIR-domain-containing adapter-inducing interferon-β (TRIF) adaptor and induces TBK1/IKKε-dependent IRF3 phosphorylation. All of these processes culminate in the transcriptional induction of Type I and III IFN genes and the production of these IFNs (Figure 1).

Newly synthesized Type I and III IFNs are secreted and activate downstream signaling in both autocrine- and paracrine-dependent manners. These two classes of IFNs bind to different membrane receptors. Type I IFNs bind to interferon alpha and beta receptor subunits 1 and 2 (IFNAR1 and IFNAR2), whereas Type III IFNs utilize the interferon lambda receptor 1 (IFNLR1) and interleukin 10 receptor subunit beta (IL10-RB) [8]. Once these receptors bind to their ligands, conformational changes recruit intracellular kinases that subsequently phosphorylate signal transducer and activator of transcription 1 (STAT1) and signal transducer and activator of transcription 2 (STAT2). Phosphorylated STAT1 and STAT2 associate with interferon regulator factor 9 (IRF9) to form the ISGF3 complex that translocates to the nucleus and activates hundreds of IFN stimulated genes (ISGs) to establish a cellular antiviral state.

PRR genes encode several splice variants to regulate their functions. A splice variant of *RIG-I* lacking exon 2 has been reported to be expressed along with the full-length isoform after IFN-β treatment and *Sendai virus* (SeV) infection [9]. This splice variant has a deletion in its N-terminal CARD domain, and this deletion prevents RIG-I ubiquitination by tripartite motif containing 25 (TRIM25), a pre-requisite for RIG-I activation. As a result, this splice variant was shown to act as a dominant negative form of RIG-I. Ectopic expression of this *RIG-I* splice variant inhibits SeV-induced IFN-β transcription. TLR3 was shown to have several isoforms [10,11]. An isoform that lacks the trans-membrane domain and most of the original intracellular TIR domain plays an inhibitory role in the IFN response [10]. The negative regulatory effects of this TLR3 isoform may be caused by competition for ligand binding because this TLR3 isoform has dsRNA binding sites, whereas it lacks the cytoplasmic TIR domain required for signal transduction. These PRR isoforms suggest the existence of a negative feedback loop that fine-tunes the antiviral IFN response.

Key signaling effector proteins downstream to the viral sensors in Type I and Type III responses express various AS isoforms as well, and many of them act in dominant negative fashion. Multiple isoforms of MAVS were observed: MAVS 1a, 1b, and 1c [12]. MAVS 1a is produced from exon 2 skipping and encodes a truncated MAVS due to a pre-mature stop codon. This truncated protein has an intact N-terminal CARD domain, and its overexpression blocks IFN-β transcription, presumably by sequestering TNF receptor-associated factor 2 (TRAF2) proteins. MAVS1b, lacking exon3, also encodes a truncated protein by a premature stop codon due to the frameshift. However, this MAVS1b is able to activate IFN-β transcription and inhibit *vesicular stomatitis*
*virus* (VSV) replication, suggesting a bidirectional mechanism by which MAVS activity is regulated. Additionally, STING, the cGAS downstream effector, has a spliced isoform, termed MRP [13]. As compared with STING, *MRP* does not contain exon 7 and thus does not have a C-terminal TBK1 interacting domain. It was shown that MRP can dimerize with STING and block STING-TBK1 interaction. This interference in STING-TBK association explains why MRP inhibits STING-mediated IFN-β transcription. In line with this finding, MRP knock-down reduces VSV replication, presumably by de-repressing host IFN responses. Intriguingly, although MRP blocks STING-mediated IFN signaling pathway induced by SeV infection, MRP enhances the *herpes simplex virus type 1* (HSV-1)-induced IFN response. Thus, it appears that MRP plays different roles in response to RNA and DNA virus infections. TRIF is a critical adaptor for the TLR3-initiated signaling pathway. A splice variant that lacks central TIR domain, termed TRIS, was observed in a wide spectrum of cell lines [14]. Previous studies have demonstrated that TRIF is associated with TLR3 through their respective TIR domains [15]; therefore, a TIR-deficient TRIS would be expected to act as an inhibitor of the TLR3-mediated signaling. However, TRIS overexpression, though to a lesser degree than TRIF, activates IFN-β transcription and knock-down of TRIS reduced poly (I:C)-induced IFN-β transcription. These results suggest a surprising, yet nonredundant, role for TRIS in TLR3-mediated signaling. Tumor necrosis factor receptor-associated factor 3 (TRAF3) is an accessory protein in RIG-I-MAVS pathways and undergoes AS in T-cells [16]. This exon 8 skipping event in TRAF3 is primarily mediated by CUGBP Elav-Like Family Member 2 (CELF2) and heterogeneous nuclear ribonucleoprotein C (hnRNP C) proteins [17]. Nonetheless, the role of this AS event in host antiviral response remains to be determined.

Novel spliced isoforms of TBK1 and IKKε have also been identified to play negative regulatory roles during the IFN response. *TBK1s*, a TBK1 spliced transcript variant, lacks exon 3–6, which encodes a serine/threonine kinase domain mediating IRF3 and IRF7 phosphorylation. Additional functional and biochemical assays show that TBK1s inhibits IFN-β transcription by blocking the interaction between RIG-I and MAVS [18]. Intriguingly, TBK1s is not abundantly expressed in uninfected cells. Upon SeV infection, particularly at later time points, TBK1s expression becomes more prominent. This delayed upregulation suggests that the cells have evolved a strategy to negatively regulate IFN activation once a viral infection is cleared. Additionally, a spliced isoform is observed upon *Influenza A* virus (IAV) infection, but its functional significance remains to be characterized [19]. Regarding *IKKε*, this gene expresses two spliced variants, IKKε sv1 and IKKε sv2, that differ in the carboxyl regions, as compared with full-length IKKε [20]. Both IKKε sv1 and sv2 form dimers with full-length IKKε and inhibit full-length IKKε-induced IRF3 signaling, including its role in promoting antiviral activity. Interestingly, *Dengue virus* (DENV) infection was observed to upregulate the expression of these two isoforms [21], suggesting that this flavivirus has evolved the capacity to interfere with innate immunity by regulating AS.

Multiple isoforms of IRF3 and IRF7 have been characterized in mammals. *IRF3a* is an *IRF3* AS variant [22,23] that utilizes an alternative exon 3a and produces an N-terminal truncated protein due to the use of a different start codon. IRF3a does not have a functional DNA binding domain and, therefore, fails to bind to IFN-β promoter. Therefore, IRF3a inhibits IRF3 transcriptional activity [22]. The second spliced isoform, IRF3-CL, is a transcript derived from an alternative 3′ splice site, 16 nucleotides upstream of exon 7 of the major *IRF3* transcript [24]. IRF3-CL shares the N-terminal region with IRF3, but differs at the C-termini. This isoform negatively regulates IRF3 activity and is expressed ubiquitously. By contrast, expression of IRF3-nirs3 is limited to specific tissues [25]. This isoform appears to be expressed in human hepatocellular carcinoma cells, but not in primary human hepatocytes. *IRF-nirs3* transcripts do not contain exon 6, and this exclusion results in a protein that lacks the central IRF association domain, which is critical for its homodimerization or heterodimerization with IRF3 or other IRFs. As expected, the overexpression of IRF3-nirs3 repressed IFN-β transcription and facilitated viral replication [25]. Additional IRF3 spliced isoforms are identified with varying degrees of capabilities in inhibiting IRF3-mediated IFN-β transcriptional activation [26]. Lastly, heterogeneous nuclear ribonucleoprotein A1(hnRNPA1) and serine- and arginine-rich splicing factor 1 (SRSF1) were shown to promote the inclusion of exon 2 and exon 3 of *IRF3* and to generate the full-length *IRF3* that is required for IFN-β transcriptional activation [27]. Depletion of either hnRNPA1 or SRSF1 causes a reduction in poly (I:C) induced IFN-β activation. More recently, IRF7 expression was shown to be regulated by the intron retention mechanism through the BUD13 protein [28]. BUD13 represses intron 4 retention in the *IRF7* transcript. As a result, a mature *IRF7* transcript is produced, and the IRF7 protein is translated to support the IFN response. In support of this observation, the knock-down of BUD13 increases intron retention of the *IRF7* transcript, which appears to be degraded via nonsense-mediated decay (NMD). Consequently, the IRF7 protein level is reduced to facilitate viral replication [28]. Several other *IRF7* transcript variants have been reported, and some may be induced by *respiratory syncytial virus* (RSV) infection [29,30].

Most Type I IFN genes are intron-less, whereas Type III IFN genes usually have multiple introns, suggesting a potential regulatory mechanism by AS. A recently discovered IFNL4, a Type III IFN, is encoded by a gene comprising five exons, and several transcript variants have been observed [31]. Functional characterization shows that full-length IFNL4 isoforms, but not shorter ones, exhibit antiviral activity [32]. Surprisingly, genetic variants in exon 1 that negatively correlate with the expression of functional IFNL4 are associated with Hepatitis C virus (HCV) clearance [31,33].

Type I and III IFN proteins exert their functions (e.g., triggering the production of antiviral ISGs) by binding to their respective receptors, which are expressed as various isoforms. IFNAR1 and IFNAR2 form a receptor complex for Type I IFNs, and IFNAR2 produces three mRNA AS variants, including two membrane-bound isoforms (IFNAR2b and 2c) and a soluble isoform (IFNAR2a) [34]. Transfection of human *IFNAR1* and *IFNAR2c*, but not *IFNAR2b*, reconstituted the antiviral IFN response [35]. This is consistent with the data that IFNAR2b may act as a dominant, negative regulator of IFN responses [36]. Multiple splice variants of IFNLR1, with which IL-10RB forms the Type III IFN receptor, have been described in human cells [37,38,39]. Membrane-bound IFNLR1 is a functional receptor subunit, whereas a soluble spliced isoform, which lacks exon 6 encoding transmembrane domain, serves as a dominant negative form. The addition of recombinant soluble IFNLR1 reduced Type III IFNs-induced ISG transcription in *peripheral blood mononuclear cells* (PBMC) and in Huh7.5 cells [40].

Following the binding of Type I and III IFNs to their receptors, phosphorylated STAT1 and STAT2 ultimately translocate to the nucleus driving ISG expression. STAT1 has two isoforms [41], alpha and beta, which differ at the C-terminal trans-activation domain. Initially, STAT1 alpha was considered to be the only functional isoform, and STAT1 beta presumably acts as a dominant negative regulator [42,43]. Nevertheless, recent studies suggest that STAT1 alpha and beta activate an overlapping, but non-redundant, set of genes that are important in regulating immunity [44]. In addition to these two isoforms, *Epstein–Barr virus* (EBV) SM proteins associate with the host splicing factor SRSF3 and promote the usage of a cryptic 5′ splice site, generating the *STAT1 alpha**′* transcript variant [45,46]. The role of the *STAT1 alpha**′* transcript, and whether or not it is translated, is yet unclear. Given the importance of STAT1 and STAT2 in driving ISG expression for antiviral state establishment, aberrant splicing of these genes has been linked with impaired immunity and severe viral illness [47,48,49]. For example, homozygous mutation leading to *STAT1* exon 3 skipping results in its reduced expression and phosphorylation. Patients homozygous for this mutation display profound susceptibility to infection [49]. A mutation in intron 4 of *STAT2* causes aberrant splicing and probably results in NMD. STAT2 protein expression is not detectable in homozygous patient cells, and the exogenous expression of STAT2 rescues the phenotype and induces an antiviral state [48].

Evidence showing that ISG function is regulated by AS is starting to emerge. OAS1 is a key component in the RNaseL 2-5A antiviral system. A recent report shows that the *Oas1g* (a mouse homolog of human *OAS1*) gene has an alternative 5′ splice site in the intron between exon 3 and exon 4, and the use of this alternative 5′ splice leads to a non-functional mRNA variant that is destined for degradation [50]. Interestingly, the removal of this alternative 5′ splice site increases *Oas1g* expression and inhibits viral infection. *MxA* is another well-known ISG that restricts various viruses. Intriguingly, HSV-1 viral infection induces the production of varMxA [51]. This transcript has exons 14–16 deleted and encodes a protein that supports HSV-1 replication. The appearance of *MxA* exon exclusion isoforms was also observed in DENV-infected cells [21]. Its regulatory function on DENV replication awaits further investigation.

In summary, various isoforms of key components in Type I and Type III IFN responses are expressed to regulate innate immunity in the host response (Figure 1). It is intriguing that most AS events downregulate the antiviral response, suggesting that posttranscriptional regulation works to balance the transcriptional upregulation of the antiviral state. This would imply that defects in the post-transcriptional regulation of the antiviral innate immune response would lead to autoimmunity and pathological inflammatory conditions.

## 3. Other Innate Immune Pathways Impacted by Alternative RNA Splicing

The promyelocytic leukemia (PML) protein, a member of the TRIM protein family, is a key component of structures known as PML nuclear bodies that have an important role in innate immune signaling [52,53]. The *PML* gene consists of nine exons and undergoes extensive AS, generating several transcript variants [54]. These isoforms share amino-terminal regions but differ at the C-terminus. Importantly, they appear to have differential roles in modulating the IFN response. PML isoform IV has been reported to enhance the activity of IRF3, thereby participating in IFN-β production during VSV infection [55]. In line with this finding, the overexpression of PML isoform IV is sufficient to suppress DENV replication [56]. Similarly, PML isoform II promotes IFN-β activation [57] and achieves this enhancement by associating with various transcriptional complexes. Depletion of PML isoform II reduced IRF3 and STAT1 recruitment to the IFN-β promoter and ISRE elements, respectively. By contrast, unlike PML isoform II, the knock-down of PML isoform V did not have an impact on poly (I:C)-triggered IFN-β activation, suggesting that PML isoform V is not required for this regulation of the IFN response [57]. Interestingly, *herpes simplex virus type 2* (HSV-2) infection caused a switch of PML isoform II to PML isoform IV by increasing the use of intron 7a through viral ICP27 proteins [58]. This is well in line with a viral strategy to antagonize the host IFN response for viral replication since isoform II, as well as isoform IV, promotes IFN-β activation. The restoration of PML isoform II in PML-knocked-down cells, however, facilitates HSV2 replication. The depletion of PML isoform II by siRNA reduced HSV2 infectivity, suggesting PML isoform II is a pro-HSV2 factor. These results suggest a complex and perhaps paradoxical role of PML in host–virus interactions.

Zinc finger protein (ZFR) participates in several cellular functions and is a potent splicing modulator. ZFR controls IFN signaling by preventing aberrant splicing and the nonsense-mediated decay of histone variant *macroH2A1* mRNAs [59]. In ZFR-expressing cells, ZFR promotes the usage of *macroH2A1* exon 6a, leading to the production of full-length macroH2A1, which represses the IFN-β promoter and prevents transcriptional activation. In ZFR-depleted cells, mutually exclusive usage of exon 6b results in a spliced transcript destined for NMD. As a consequence, the IFN-β promoter is released from repression and can be accessible to transcription factors for gene expression. Consistently, either knock-down ZFR or macroH2A1 enhances IFN-β transcription. Furthermore, ZFR depletion restricts viral replication [59].

hnRNP M belongs to the family of ubiquitously expressed heterogeneous nuclear ribonucleoproteins (hnRNPs) and impacts pre-mRNA processing and several other aspects of mRNA metabolism and transport. Recently, hnRNP M was shown to possess immune-suppressing capability via distinct mechanisms. Firstly, this protein interacts with RIG-I to impair immune sensing [60]. Moreover, hnRNP M promotes intron retention to reduce IL-6 transcript abundance. Overall, as a consequence, the depletion of hnRNP M dampens host immunity and facilitates viral replication [61].

## 4. Alternative Splicing Regulates Host Cell Death Pathways Activated during Viral Infection

Several cell death programs have been described, and the molecular mechanisms of these programs are overlapping, yet quite divergent [62]. Here, we discuss AS regulation of apoptosis, necroptosis, and pyroptosis in the context of host–virus interactions (Figure 2). Viruses interact extensively with cellular intrinsic and extrinsic apoptotic pathways [63,64], and the spliced isoforms of apoptotic factors can play a key role in determining cell fate [65,66]. Apoptosis is generally conceived as a non-inflammatory type of programmed cell death, characterized by morphological changes, including cell shrinkage, nuclear condensation, and plasma membrane blebbing. Intrinsic apoptosis is primarily controlled at the mitochondria. The disturbance of intracellular homeostasis (e.g., DNA damage or oxidative stress) and pro-apoptotic stimuli results in the induction of mitochondrial outer membrane permeabilization (MOMP) by the proteins BAX or BAK. In the absence of apoptotic stimuli, these proteins are sequestered in an inactive state by anti-apoptotic members of the BCL2 family of proteins. MOMP releases cytochrome c into the cytoplasm and triggers the formation of the apoptosome protein complex containing apoptotic protease activating factor 1 (APAF1) and caspase 9, a member of cellular cysteine-aspartic proteases. Activated caspase 9 then cleaves caspases 3 and 7, which triggers a pathway leading to cell death. Interestingly, viral infection activates non-transcriptional IRF3–Bax interactions and causes MOMP and cell death [67,68]. Viral infection can also trigger extrinsic apoptosis [63]. The extrinsic pathway is primarily initiated by binding of ligands to various death receptors, activating caspase 8, and leading to the activation of caspase 3 and apoptosis. A key host factor in regulating this pathway is the FLICE-like inhibitory protein (cFLIP). Human *cFLIP* has 14 exons and gives rise to three major isoforms cFLIP_L_, cFLIP_S,_ and cFLIP_R_ [69]. Both cFLIP_S_ and cFLIP_R_ are shown to block apoptosis [69,70,71], while cFLIP_L_’s role in regulating apoptosis is dependent on its local availability and concentration [72]. In addition to regulating apoptosis, cFLIP_L_ has recently been shown to inhibit IRF3 and IRF7 transcriptional activation, suggesting the critical role of cFLIP in the host’s response to viral infection [73,74]. Splicing regulatory mechanisms by which various cFLIP isoforms are generated are currently being revealed. A 21-bp insertion in the 3′UTR of the fifth exon in the *cflar* gene causes preferential splicing to cFLIP_L_ [75]. Intriguingly, this effect was mainly observed in liver tissue, suggesting that additional tissue-specific trans-acting factors are important to modulate this splicing process. In addition, a single nucleotide polymorphism is reported to dictate whether cFLIP_R_ or cFLIP_S_ is produced [76]. It is important to note that though cFLIP_S_ inhibits apoptosis, it promotes another programmed cell death, termed necroptosis [70].

Necroptosis is an inflammatory type of cell death, characterized by cell swelling, loss of plasma membrane permeability, and the release of cytosolic contents into the extracellular space [77]. Some viral infections trigger necroptosis through membrane-bound receptors (e.g., TLR3 [78,79]) or cytosolic sensors (e.g., ZBP1 [80,81,82]), and culminate with the activation and phosphorylation of mixed lineage kinase domain-like protein (MLKL), which forms a homotrimeric complex that translocates to the plasma membrane, where it forms a pore and induces cell lysis [83,84]. MLKL has two isoforms, MLKL1 and MLKL2 [85]; MLKL2 is a more potent necroptosis inducer than MLKL1 [86]. This increase in activity can be attributed to the altered domain structure of MLKL2. *MLKL2* lacks exons 4-8 and, thus, MLKL2 does not have most of the C-terminal pseudokinase domain that is believed to serve as a suppressing function. Additional key components in the necroptosis pathway include receptor-interacting serine/threonine-protein kinase 1 (RIPK1) and receptor-interacting serine/threonine-protein kinase 3 (RIPK3), and both genes have been reported to encode transcript variants. CRISPR whole-genome screening identified PTBP1 as a novel regulator of *RIPK1* splicing in necroptosis [87]. PTBP1 suppresses an alternative exon inclusion between the canonical exons 4 and 5 and promotes full-length RIPK1 protein expression for cell death induction. In line with PTBP1-mediated exon skipping, a signature CU-rich tract in the intron adjacent to the 3′ splice site was identified [87]. Lastly, *RIPK3* has two splice variants, *RIPK3 beta* and *RIPK3 gamma*, both of which appear to suppress cell death [88].

Pyroptosis, which is also an inflammatory form of cell death, is induced by inflammasome activation and is critical for the antiviral response [89,90]. This death pathway is initiated by the detection of PAMPs or danger-associated molecular patterns (DAMPs) by inflammasome proteins, most of which are members of the Nod-like receptor (NLR) family [91]. Subsequently, the recruitment of adaptor apoptosis-associated speck-like protein containing a CARD (ASC) and caspase 1 forms an inflammasome protein complex. Activated caspase 1 would cleave gasdermin D (GSDMD), releasing the GSDMD-N domain. The GSDMD-N domain translocates to the plasma membrane and oligomerizes to generate membrane pores. This pore formation disrupts the osmotic potential, resulting in cell swelling and eventual lysis. In addition, active caspase-1 processes pro-IL-1β and pro-IL-18 to bioactive forms, promoting inflammation. Among many NLRs, NLR family pyrin domain containing 3 (NLRP3) is important in antiviral response [89], and it was recently shown to be regulated at the splicing level. A novel *NLRP3* spliced variant that lacks exon 5 encodes a fraction of the LRR domain [92]. Deletion of this region abolishes its interaction with the NIMA-related kinase 7 (NEK7) protein, the binding of which is a prerequisite of NLRP3 activation, and thus makes NLRP3 ∆exon5 inactive. Another critical component in the inflammasome is the adaptor ASC, composed of an N-terminal PYD domain for association with the NLR proteins, a linker region, and a C-terminal CARD domain for interaction with the caspase proteins. A spliced variant of *ASC-b* lacks exon 2, encoding the linker, and is able to activate NLRP3 inflammasome [93]. In line with this, a patient harboring this *ASC* exon 2 deletion exhibits a higher level of IL-1β protein in the serum [94]. Another spliced isoform ASC-c has a deletion in the PYD domain. It acts as a dominant negative regulator and reduces NLRP3 activation [93].

## 5. Conclusions

Viral infection triggers myriads of cellular events in the host. Most earlier studies have focused on the role of transcription in setting up the cellular antiviral response. In this review, we discuss the underexplored role of AS in regulating the host response during viral infection. Most AS regulatory events discovered to date appear to negatively modulate the antiviral response. This would imply that defects in the post-transcriptional regulation of the antiviral innate immune response would lead to autoimmunity and pathological inflammatory conditions. With the advent of next-generation sequencing, novel discoveries have been made to delineate how the host splicing machinery is modulated during viral infection [95,96]. It is evident that AS plays critical roles in the regulation of a productive innate immune response; however, the functional significance of many novel AS isoforms and the regulatory mechanisms by which these spliced variants are generated remain incompletely understood. Further investigation should explore AS as an important layer of regulation of virus–host interactions and potentially identify novel targets for therapeutic development to treat infectious diseases.

## Figures and Tables

**Figure 1 cells-10-01720-f001:**
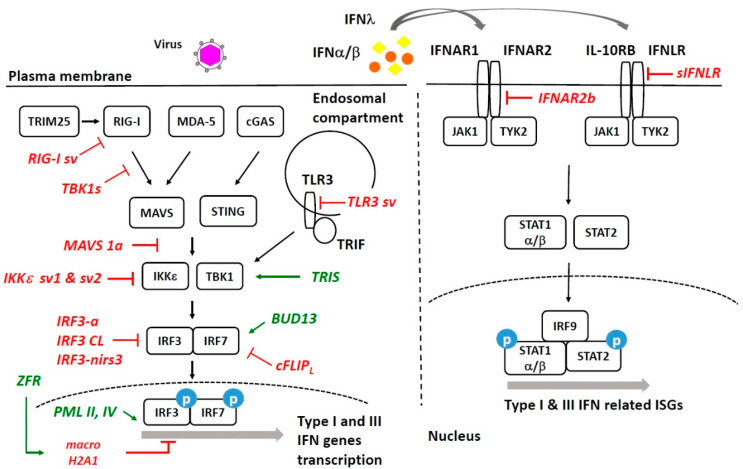
Alternative splicing in the host Type I and Type III IFN response. AS isoforms that upregulate the antiviral response are shown in green, and those that downregulate the antiviral response are shown in red.

**Figure 2 cells-10-01720-f002:**
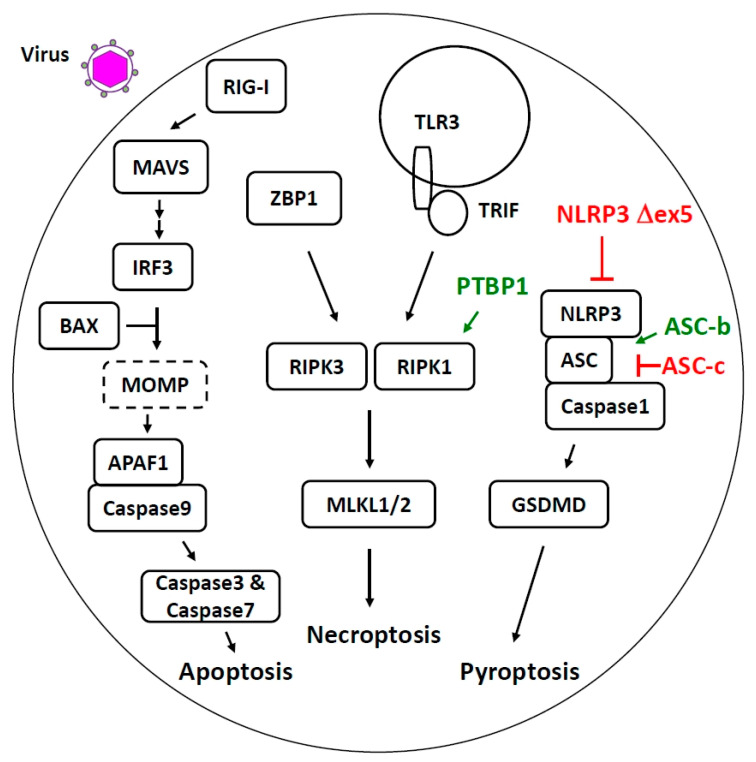
Alternative splicing regulates host cell death pathways activated during viral infection.

## Data Availability

Not applicable.

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
