# Peer review of "Role of Alternative Splicing in Regulating Host Response to Viral Infection"

_cells, 2021, doi:10.3390/cells10071720_

Round 1

Reviewer 1 Report

This review by Liao and Garcia-Blanco is shedding light on a layer of regulation of innate immune responses to viruses that is often dismissed but important: alternative splicing. This current manuscript presents a compelling collection of studies, is well-written and really interesting to read.

I only have a few minor suggestions:

A second figure depicting the cell death pathways would be much appreciated.

line 36: Info on RNAPOlI and III alternative splicing?

Paragraph 2, the names of the proteins are written in a different font, it could benefit from subheadings, like Pattern recognitions receptors, Signalling transduction factors, Transcription factors, etc.

line 53: differentiate between RIG-I and MDA5 regarding the ligand, RIG-I can also efficiently bind di-phosphorylated, whereas phosphorylation doesn’t seem to be a prerequisite for MDA5 activation

line 62: why is that ‘consequently’?

line 69: rather transcriptional induction than activation

Figure 1: The virus may rather be symbolized by a viral capsid than a pink splash. Please add an arrow from intracellular IFNs to the outside or remove intracellular IFNs.

line 77: an additional space after subunit 1.

line 89: add a space after 25

line 107: ‘a complicated mechanism’ is a bit vague, is there a more definitive description which could be chosen? Why is it complicated?

line 107: I would not call STING a cGas adaptor as both proteins do not interact within the cascade, but rather rely on a second messenger to communicate.

line 155: “It appear…” would be more suitable as a sentence a bit above, before describing the different functions. Or as Thus, it appears that…

line 129: remove this before TBKs

line 131: brake? Maybe negatively regulate would be better here?

line 142: There is an extra space before IRF3a

line 148: Maybe rather: …. expression of IRF3-nirs3 is limited to specific tissues.

line 150: This sounds like the exon 6 need to be excluded from IRF3a-nirs, not that it has it already excluded, please rephrase

line 161-164: IRF7 instead of Irf7

line 165: A bit vague, they could be listed, any functions known?

line 179-181: That was said in a paragraph above, could be shortened

line 198ff: Usually abbreviated as OAS1

line 204: Restricts various RNA viruses, and then HSV-1 is brought up as an example sounds a bit inconsistent

line 209: A reference to Figure 1 could have been made earlier.

line 216: is a key component

line 239-245: Different font

line 239: What are ‘normal’ conditions

line 243: Please rephrase ‘IFNb promotor is derepressed’

line 255-257: Could be rephrased to less vague statements such as ‘overlapping yet quite divergent’.

line 257: Viruses

line 263: There is an extra comma before DNA damage

line 283:  are currently being revealed (?)

line 291: Some viral infections triggers ….

line 311: I would not call them scaffold proteins, please rephrase.

line 319: There is no need to mention that it is a key player

line 331: There is an unnecessary dash after conclusion

line 344: Maybe rather, AS as an important layer of regulation of virus-host interactions.

line 347: The funding statement and conflict of interest statement are missing.

Author Response

Thank you for the detailed review and positive comments - all changes have been addressed.

Reviewer 2 Report

In this short but comprehensive manuscript Liao and Garcia-Blanco review the role of alternative splicing in regulating antiviral IFN activation and signaling pathways as well as different cell death mechanisms. The authors aptly write about most descriptions of alternative splicing of factors involved in these pathways. As such, this is a valuable review that highlights the understudied role of alternative splicing in antiviral pathways.

However, this review would benefit from the additional discussion of a few points:

  1. TRAF3, an accessory protein of RLR signaling, undergoes CELF2-mediated alternative splicing in T cells. While this hasn’t been shown to directly regulate viral infection or antiviral immunity, this finding warrants brief discussion (PMID: 28031331).

  1. While most IFN transcripts do not seem to be regulated by splicing, alternative splicing of IFNL4 may work to suppress expression of this gene, with host genetic variation being implicated. It would be important to discuss this. (PMID: 31846709, 27799623, 23291588).

  1. One critical aspect that was generally lacking from the review was a discussion of the mechanisms underlying alternative splicing during immune activation/viral infection. WHY does induction of many genes in these pathways following stimulus lead to alternative splicing and feedback loops? Thus, this manuscript and the readers thereof would greatly benefit from a section detailing modes of alternative splicing regulation such as differential expression of splicing factors, post-translational modifications that affect splicing factor function, and the integration of cellular signaling events into splicing regulation.

Minor points:

  1. Line 257 – typo in the word “Vruses”

  1. Several protein names are misformatted across the manuscript, appearing in different fonts and italics.

Author Response

Thank you for your thoughtful comments - please see our response to the editor.
